# Literature Review of Accreditation Systems in Higher Education

Nelson Duarte [1] and Ricardo Vardasca [1,2,*]

1 ISLA Santarém, 2000-241 Santarém, Portugal; nelson.duarte@islasantarem.pt
2 INEGI, Universidade do Porto, 4200-465 Porto, Portugal
* Correspondence: ricardo.vardasca@islasantarem.pt

**Abstract:** This study investigates the accreditation processes in higher education across various countries, focusing on the time and bureaucratic burden associated with accrediting new courses. The aim is to identify strategies to accelerate the accreditation process for new courses in higher education institutions. A comprehensive literature review was conducted to achieve this objective, examining the accreditation processes in Portugal, Spain, the United States, France, China, Japan, Sweden, the United Kingdom, India, and Germany. The study's key findings revealed that the accreditation process is generally efficient in most countries, with courses receiving accreditation within a reasonable timeframe. However, the process can be more complex and time-consuming for institutions seeking accreditation for the first time or offering new or innovative courses. Institutions must meet all established criteria and promptly provide all required documentation to expedite the accreditation process. The implications of these findings suggest that higher education institutions should collaborate closely with relevant accrediting agencies to ensure a streamlined accreditation process. Institutions should also consider agency requirements and course specialization when developing new courses. Furthermore, governments play a crucial role in promoting transparency and competition among higher education institutions, which can lead to enhanced quality assurance and increased customer satisfaction in the education sector.

**Keywords:** academic courses; degree accreditation; higher education; management; quality system

## 1. Introduction

Accreditation in higher education is a key process that attempts to ensure the quality of institutions and their respective programs. It serves as a critical mechanism to ensure that educational providers maintain standards that meet the expectations of the educational community and society at large. Despite its universal importance, the approach to accreditation varies significantly across different regions and even within the same country, reflecting the complexity and diversity of higher education systems worldwide. This study focuses on institutional and programmatic accreditation, providing a comprehensive overview of different accreditation processes in selected countries worldwide. The decision to include both forms of accreditation arises from the recognition that quality assurance in higher education operates at multiple levels. While institutional accreditation assesses the overall quality of an institution, programmatic accreditation evaluates specific programs within institutions, offering a more detailed understanding of quality assurance mechanisms. The study encompasses ten countries—Portugal, Spain, the United States, France, China, Japan, Sweden, the United Kingdom, India, and Germany. The selection of these countries was guided by their representation of different higher education systems, cultural contexts, and accreditation practices, thereby enriching the comparison and analysis of accreditation processes. Utilizing a systematic review of existing literature, this study aims to map the accreditation landscape in the chosen countries and analyze the time and bureaucratic burden involved in these processes. The paper aims to propose strategies to

expedite the accreditation process of new programs or courses, thereby contributing to the ongoing discussion on improving efficiency in quality assurance. The paper is organized as follows: Section 2 presents the data collection and analysis methods. The results and discussion, which delve into the specifics of each country's accreditation process, are presented in Section 3. Finally, Section 4 offers conclusions from the study and directions for future research.

This study contributes to the existing body of knowledge by offering a comparative perspective on accreditation processes, highlighting the commonalities and differences across countries and proposing measures to improve efficiency in accreditation. This understanding will be crucial for policymakers, educational institutions, and accrediting agencies seeking to refine their accreditation processes and enhance the quality of higher education.

## 2. Materials and Methods

This study used a systematic review as a research method and process to identify and analyze key research findings. This section describes the systematic literature review approach to analyzing the accreditation processes in higher education across various countries. The selection criteria for articles were guided by a conceptual framework that focused on the following key dimensions: accreditation processes, quality assurance systems, the role of governments, and the challenges and opportunities in accrediting new courses [1].

Regional selection was used to provide diverse representation of higher education systems worldwide. Consequently, the selected countries were from different continents, including Europe, North America and Asia. These countries were Portugal, Spain, the United States, France, China, Japan, Sweden, the United Kingdom, India, and Germany. By examining a broad range of countries, we aimed to identify common themes and unique aspects of the accreditation process, which could inform strategies for accelerating the accreditation of new courses in higher education institutions globally [2]. Academic databases such as Scopus, Web of Science, and Google Scholar were searched using keywords related to accreditation, quality assurance, higher education, and new courses to identify relevant articles. The reference lists of identified articles were also reviewed to find additional relevant studies. The articles were then screened for relevance based on the title and abstract, which was followed by a full-text assessment. Articles that met the selection criteria were included in the review and analyzed according to the dimensions of the conceptual framework. To better integrate Harvey and Green's framework defining quality in higher education within the analysis, their five dimensions of quality—exceptional, perfection (or consistency), fitness for purpose, value for money, and transformative—were applied to the accreditation processes and quality assurance systems in the selected countries. Throughout the Results and Discussion section, connections are drawn between the accreditation processes and the quality dimensions defined by Harvey and Green. For example, when discussing the role of governments in promoting quality education, we can relate this to the "value for money" and "fitness for purpose" dimensions by explaining how governments set standards to ensure that higher education institutions provide valuable and purposeful education to students. Similarly, when exploring the challenges and opportunities associated with accrediting new courses, we can connect this to the "exceptional" and "transformative" dimensions. It was possible to describe how innovative courses may face more complex accreditation processes because they push the boundaries of traditional education, thus aiming for exceptional and transformative outcomes.

The "perfection (or consistency)" dimension is also discussed in the context of accreditation processes to strengthen the connection between Harvey and Green's framework and the analysis. For instance, it is discussed how accrediting agencies evaluate higher education courses based on their consistency in delivering high-quality education, adhering to established standards, and continuously improving their processes. By explicitly linking the accreditation processes and quality assurance systems to Harvey and Green's framework throughout the analysis, we aim to provide a more cohesive understanding of quality

in higher education and how it is maintained and promoted through accreditation [3]. The complete list of reference articles is provided in alphabetical order.

### 2.1. Questions to Investigate

This study aims to explore the patterns, similarities, and differences across countries. The following questions were considered:

What are the common criteria for accreditation across countries?

How does the length of the accreditation period compare between countries?

Are there common challenges faced by institutions seeking accreditation for the first time or offering new or innovative courses?

### 2.2. Quality Assurance Methods in Higher Education

This section emphasizes the significance of implementing precise and effective methods for quality assurance in higher education and the importance of communicating these results through various mediums. The techniques employed across Europe, the United States, and Asia were researched based on their diverse approaches to higher education quality assurance, significant influence on global education policies, and varied cultural and administrative contexts. This diverse selection provides a comprehensive overview of global trends and methodologies.

Accreditation is a method predominantly used in the United States. The work of Massy [4] underscores the necessity for more sophisticated quality assurance methods in higher education. These are categorized into three main types: accreditation, evaluation, and peer-reviewed quality audits. Figure 1 illustrates these categories of quality.

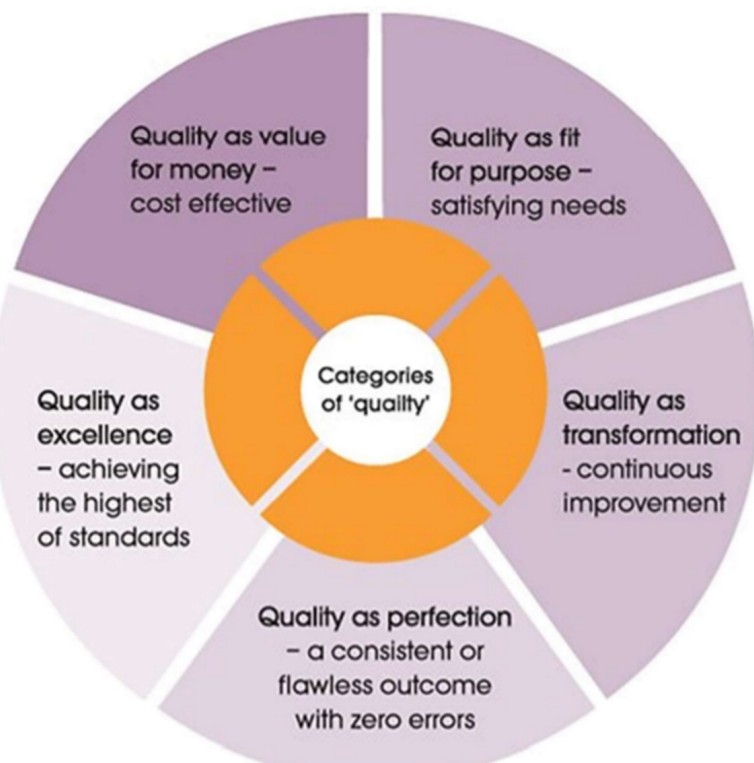

**Figure 1.** Illustration of Harvey and Green's five ways of thinking.

Evaluation refers to the standards of European education, while peer-reviewed quality audits are a blend of performance indicators, self-study, and review processes. These different methods collectively contribute to enhancing quality in higher education institutions.

### *2.3. Process Review and Quality Auditing*

Quality audit, in contrast to process review, focuses on analyzing the internal quality and adopting improvements in these processes. Several studies in various countries have concluded that inducing capacity building can significantly improve learning and teaching methods in higher education institutions [4].

Many Higher Education Institutions (HEI) have used the scorecard method applied to business excellence to achieve excellence status, allowing them to improve their processes through organizational excellence acquired through crucial elements, organizational learning, and significant improvements in the processes used [5]. The quality of the processes must be ensured internally by the institutions and must be continuous to improve the institution's quality.

### *2.4. Evaluation and Quality*

The studies presented in this section contextualize quality and evaluation in Europe, China, Japan, India, and the United States. To this end, a conceptual mapping of the quality scenario involving the HEIs in all these countries is provided.

### 2.4.1. In the United States

Accreditation in the USA is based on the results achieved in a public process to show whether the higher education institutions respect the defined quality objectives. Innovative methodologies in teaching can be reflected in better outcomes in course accreditation [6]. This type of accreditation in the United States is unique to education. It is common in undergraduate education, where a comparison is made of the performance of educational institutions against defined metrics determined by the government or accrediting agency. This process will evaluate if the proposed objectives for the degree to be accredited have been met. This type of certification must be done by agencies external to the HEI. This accreditation is completed in stages and will ensure that the minimum requirements related to quality are met. The results must be disclosed and published for the institution to receive the certification [7].

### 2.4.2. In Germany

Studies have been selected that address the theme's evolution between 1994 and 2015 [8–11]. According to Geisinger [8], the analysis of HE assessment has highlighted a deplorable assessment situation in Germany. This situation decreased public investment in the government's scientific and educational funds. Reports and surveys exposed this reality in the 1980s and 1990s. German students were compared to students in the rest of Europe. It was determined that German students needed three to five additional semesters to obtain their degree [8].

The creation of the Foundation of the Center for the Development of Higher Education (Zentrum für Hoch-Schulentwicklung) in Giitersloh was an initiative to promote the quality of HE at German universities through internal and external evaluations and external evaluations of teaching performance. Although some instruments were developed for course evaluation, their application, significance, and publication of comparable results were controversial [8]. There were also severe problems in establishing comparisons between the performance of the universities, which began to follow recommendations for standards suggested by the Conference of Rectors and Presidents of Universities and HEIs, in the annual teaching reports [8].

De Wit and Hunter [11] explained the complex decision-making in HE concerning legal requirements, administrative planning, and financial issues in the federal state of Germany. They analyzed the increase in HE privatization and the legal and economic problems. Introducing new financing systems based on performance indicators augurs well for primary status structure and legal management changes. These changes dramatically increased between 1975 and 2000, reaching about 1.8 million [11]. While HE vacancies have not kept up with this growth, neither has the number of teachers.

A reform of the German HE system was introduced by calling for greater competition and differentiation through deregulation, performance orientation, and performance incentives. These reforms are part of the policy of increasing the competitiveness of German HE and provide necessary measures, including public funding of HEIs and allocation of resources at the institutional and departmental levels, which are based on performance indicators [11]. Harris-Huemmert's research criticizes Germany for not having standards for selecting HE evaluators or evaluating their performance [9]. The analysis contextualized the processes of German HE evaluation through the study in the assessment of science education in Baden Württemberg and explored how experts were selected. The study describes the problems faced and finishes by showing there might be space for introducing standards for determining evaluators in the German evaluation scenario. A more recent study from Orr and Hovdhaugen [10] argues for widening access to HE as proposed by the European policy agenda through a second chance route by removing the criterion of educational attainment in secondary education as the determining factor for access. An analysis compared the approaches of similar routes in Germany, Norway, and Sweden with different forms, principles, and obligations for HEIs. Orr and Hovdhaugen [10] evaluated the impact of second-route opportunities to widen participation in HE and discussed the contribution of these measures to access and inclusive education in the country.

### 2.4.3. In the United Kingdom

The Higher Education Funding Council for England (HEFCE) structures the assessment process in England. It is separated into two assessments: one focused on teaching and the other on research. The goal is to create quality profiles based on clarity, transparency, efficiency, neutrality, coherence, credibility, continuity, and parity [12]. The methods the agencies use to evaluate higher education institutions, such as pedagogical projects, faculty, and course infrastructure, are similar to the techniques presented by Brenan and Shah [13]. Government entities, consortiums, or HEIs usually do this process. For HEIs to be held accountable, the results of the evaluations must be published for comparison with those of other HEIs. These external evaluations occur for five to ten years [4].

The concern with quality standards and assessment applied to HE has been evidenced in several approaches [14–16]. The study by Athanassopoulos and Shale [14] compares corporate performance in 45 HEIs in the United Kingdom. Government initiatives in this sector have emphasized responsibility, worth for money, and cost regulation encouraged by the concepts of new public management (NPM) and its derivation into the official weighting scheme followed by the funding agency in the sense of establishing measures and criteria for allocation of resources to universities. Quality is evaluated regarding return on investment or expenditure. The value-for-money approach in education can be associated with accountability. In public services, including education, accountability to funding and funding agencies is expected. Focusing on cost control, Athanassopoulos and Shale [14] adopted two distinct models for the definition of performance, cost minimization and result maximization. The two models can be complementary to measure the HEI result's cost efficiency. According to the proposed methodology, the study rated six universities in the United Kingdom that showed satisfactory individual performance in all tests [14]. The universities 'activity statistics have reached their total potential in defining comprehensive perceptions of execution and objective attainment concepts advised by institutional objectives. Research on total quality management (TQM) in HE from the United Kingdom [17] reports that the progress in the use of TQM has been slow, with only a few universities adopting it. The HEIs that have used a TQM method similar to their partners in the United States have shown improvements in student performance, improved services, decreased costs, customer gratification, and customer costs [18]. The authors examined how the fundamental principles and concepts of TQM could be evaluated to evaluate the quality of HEIs in various aspects of their internal processes.

The TQM principles and fundamental concepts constituted critical success factors and reflected the performance of the institutions. The business excellence model [17]

has demonstrated several advantages and has overcome shortcomings over other quality management models. A study performed using external monitoring of HE quality [15] examined the different types of external agencies and their modus operandi. Harvey criticizes using statistical indicators as assessment tools because they have their limitations as quality performance measures. His prerogative is that evaluation legitimizes the status quo and is concerned with the method, ignoring the nature and styles of learning styles. He further contests that self-evaluation forces peer review to occur. In this peer review and, in this specific case, it is a biased, distorted judgment on the search for discrepancies and personal assessment. In [15], he radically proposes that HE monitoring agencies must be concerned with quality, address implications for student learning, change from education, and move from accountability and compliance-oriented agencies to raise important issues of improving student learning.

In a study of quality and standards [19], English HE is discussed in regard to ensuring quality. The study focuses on the attributes of the English system centered on self-regulation and self-governing institutions. The analysis interrogates the process of the quality policy transition that occurred during the consolidation of the 2006–2011 evaluation cycle. The historical review performed by Quality Assurance Agency demonstrated that there was an initiative between institutional evaluations and program evaluations. The institutional assessment took place through audit and was conducted by the Higher Education Quality Council (HEQC). Program evaluations occurred through quality assessment with subjective instruments by the HEFCE. It was found that with the merger in 2006, the QAA absorbed the functions performed historically by the HEQC's predecessor agencies, HEQC and HEFCE, demonstrating great vagueness in defining systems and methods currently used to assess the guarantee of academic values in the United Kingdom. The audit processes are carried out by assessors, namely senior HEI staff instructed by the QAA, who examine documentation and cooperate with staff and learners to assess the efficiency of quality methods. Since January 2010, students have been included in the audit teams.

The evaluation of research work in the United Kingdom indicates an academic struggle for quality-guarantee methods in HEIs. The processes were perceived to monitor and control academic work related to teaching and research in the country's HEIs over the last 20 years. Academics in specific contexts have challenged and resisted discourses and positions imposed on them [20].

2.4.4. In China

The core of the Chinese higher education reform, guided and promoted directly by the government, is the development of content that raises the education quality and the capability to innovate and improve the modern socialist system of a higher system of higher education. The most visible effect of the reform has been a significant movement toward the autonomy of the HEIs themselves, manifested in initiatives in areas such as undergraduate studies, internationalization, and computerization. In addition, educational equity and the local values of higher education—to involve all the significant elements of student growth—has been prioritized to thereby develop the content of HEIs to raise education quality.

Improving the quality of education has become a comprehensive, organized project that depends on a modern university system, progressive ideas on education, a system of internal quality assurance in universities, and efficient, systematized management mechanisms that demand attention from the government, HEIs, and society. For this reason, since the early 1980s, China has conducted undergraduate assessments. In 2003, through the Education Rejuvenation Action Plan (2003–2007), the Ministry of Education proposed a five-year evaluation system of pedagogical work in the HEIs. In August 2004, it created the Center for Evaluation of Education and Teaching, directly subordinated to this Ministry. As a result, the country's evaluation of education and teaching has been pushed forward. Teaching in the country has gradually become more systematized, standardized, professionalized, and scientific. In 2007, the Ministry of Education and Finance proposed

another essential project, implemented vertically across the country, to realize educational reform in popularizing higher education. The Quality Reform Project for Teaching and Undergraduate Education Reform Project, also known as the Quality Project, was based on three other projects: the National Renowned Professor and the National Course of Excellence, effective in 2003, and the National Center for Demonstration of Experimental Teaching, effective in 2005. The Quality Project received input from central government funding of CNY 2.5 billion during the Eleventh Five-Year Plan (2006–2010).

In 2011, based on the summary of the evaluation experiences of previous years, the Ministry of Education promulgated an opinion on the undergraduate evaluation work of HEIRIs, determining that the main content of institutional evaluation had to be certification and professional assessment, international evaluation, and control of data on the basic framework of education. In early 2012, this Ministry-certified evaluation program of the news cycle published measures for the implementation of evaluation and approval of the pedagogical work of HEIs. On 16 March 2012, the Ministry published an opinion called 30 Items for Higher Education on considerably increasing quality improvement in higher education. Moreover, it determined that higher education must persist with a stable scale, qualify the structure, fortify the peculiarities, prioritize innovation, and move toward content development, having the elevation of quality at its core. It further determined that "the scale-up of higher education has as its main functions, develop higher vocational education, continuing education, master's degrees, expand private education and cooperative education" and also "look into the creation of a classification system of HEIs, elaborate classification management measures, and overcome the trend of homogenization". Innovative measures are proposed in talent training, scientific research, social service, cultural areas, and educational management. Actively cultivating skills that meet social development needs and limiting recruitment into courses with high employability are emphasized. This initiative by the Ministry of Education will benefit HEIs by respecting their realities and guiding them scientifically to raise the quality of education through personalized construction [21].

2.4.5. In Japan

In Japan, a rapidly developing economy after World War II was followed by the growth of HEIs. The development was financed by private companies, which still manage roughly 80% of HEIs. These private HEIs have a good quality of education, with the remoteness of traditional private universities tending to vary constantly among these HEIs. From early on, the Institution Approval System created by the HEIs was controlled by the Institution Approval System. In the early 1990s, the approval system was set aside, giving institutions more autonomy to structure their courses. New evaluation forms were created, making institutions more directly responsible for assuring quality. In 2001, the MEXT—Ministry of Education, Culture, Sports Science and Technology—created rigorous parameters and norms to control quality in HEIs.

Laws were created so HEIs could evaluate themselves and be evaluated by external companies. The revisions described by the OECD in Japan [22] support more autonomy and freedom in HEIs, allowing Japan to restructure its evaluation standards and education systems more efficiently. The quality control methods in Japan consist of an approval process where HEIs have to evaluate and control themselves using the CEA process and an evaluation process from the National University Corporation. This specific and rather complicated evaluation method is widely contested for various reasons. Arguably, it has never been perfected, nor is it appropriate. This is worrisome, especially in national institutions, because there are processes to prepare for the final annual, midterm, and biannual evaluations [22]. These processes are time-consuming for the faculty and administrative staff involved in these operations and often occur at the expense of studies and related research and teaching. Japan's accreditation and evaluation programs have become more rigorous to increase accountability and improve the reforms to be implemented. Therefore, whether this system will make HEIs more competitive and resilient remains to be seen.

### 2.4.6. In Sweden

The supervision-rather-than-responsibility approach has traditionally been dominant in Sweden in public administration and was implemented in HEIs until the 1980's. The assumption was that education was expected to have the exact requirements, conditions, and quality, regardless of where it was delivered. From then on, state control and supervision gradually began to give way to institutional independence and growing self-regulation [23]. With the gradual democratization of the management of higher education institutions, opportunities were given to various stakeholders to impact the progress of teaching and research. The HEI reform of 1993 allowed for this development by decentralizing the funds' studies, assignments, and internal assignments. A new university funding system was established based on achievement, performance, and student numbers to encourage intensive research, teaching, and administration processes. Every institute oversees its activities' quality and development. The "Swedish model" has developed gradually since the end of the 1980s. Its implementations are now the subject of intensive critical discussion [24], resulting in new development. Therefore, the following overview and analysis must be seen as a progressive report of the last years of the 1990s. The requirements for the institutional independence and public accounting of HEI activities have significant legitimacy in Sweden, and the debate concerns mainly the tradeoff between evaluation and auditing.

The duty of performing evaluations is assigned to the National Agency for Higher Education, which audits and assesses universities on the national evaluation of subjects and curriculum. These assessments cover teacher training, mathematics, medical training, paramedical programs, and doctoral programs in languages. The sites for evaluation, which could also have structural phenomena, are selected based on identified problems or other criteria, accreditation of specific programs and degrees at all institutions, and accreditation of colleges applying for university status. This method has proven to be one of the most effective quality-driving measures to improve college standards. It is carried out based on established criteria for all assessments. Among these criteria are the proportion of faculty with doctoral degrees, the nature and quality of research, the number and scope of advanced courses, library resources, and other facilities, with the quality audit examining the institutions' systems to ensure the quality of the research and their operations. Together, they intend to create a single system that can guarantee quality throughout the country. Currently, the form of evaluation enforcement is auditing, which is used to improve and provide ways for the government to audit the quality processes implemented in higher education institutions. It is a system controlled by the government and negotiated to promote the growth of the institutions, intending to create routines for undergraduate, graduate, administration, research, and development courses. These audits are carried out in three- and four-year cycles, and there is no link to funding. At the end of the audit, a self-assessment is performed by the institution related to the institution's quality improvement, with this being the most relevant documentation for the auditors. Then, there is a peer review in teams set up by the National Agency, each group consisting of three or four professors, academics, an individual from industry or related to government administration, and a student. All materials are analyzed, conclusions are presented, and a two-to-five-day review begins. When the reports are published, the chancellors meet with the Swedish educational institutions and their administrations to discuss what measures should be carried out according to the audit process. One year after the audit, the chancellor will again visit the institution to discuss measures for developing and publishing the report. All institutions have discussed and accepted this model [24].

### 2.4.7. In Spain

Recent academic publications have discussed the quality assurance in higher education in Spain. The topics presented in these publications are related to those in the literature in other European countries on quality assurance. Issues such as the evaluation of student training outcomes [25–29], accountability [30], skills and labor market demands, [25,27,31],

internationalization and mobility [32], educational reform [33], and teacher quality [34,35] are widely mentioned in the literature. In Spain, a frequent topic in academic papers related to higher education quality is the impact of economic factors [35]. The economic recession in Spain has restricted some of the monetary resources available to HEIs. This topic is widely covered in various papers. The commitment to establishing and improving the quality of educational systems is not exclusive to Spain. This characteristic can be observed in various European countries and has been a part of the recent adjustments driven by higher education in Europe. The European Association for Quality Assurance in Higher Education has published the Standards and Guidelines for Quality Assurance in the European Higher Education Area [36]. The European Ministers approved these rules during the Bologna Process Summit in Bergen, Norway, in 2005 [25]. The king signs legislative royal decrees and acts.

### 2.4.8. In France

In France, the government manages the model of education administered in higher education institutions. The state department responsible for higher education does not oversee it entirely. Certain institutions maintain higher education programs (industry, cultural affairs, agriculture) with specific forms of operation. The nonprofit institutions are primarily private institutions of higher education, many Catholic universities. The teachers' unions significantly impact education administration by being part of the advisory boards that advise the government. HEIs have some autonomy in the administration of their internal operations. The boards of directors have representation from teachers, students, business people in the industry, and politicians. HEI directors are eligible to serve a single five-year term. The government, specifically the ministers, appoints the HEI directors [37].

The form of official accreditation is essentially based on diplomas and degrees. An institution can access an accredited degree based on its programs or plans to offer students and the resources it intends to apply to teaching and training. This accreditation process in universities is completed every four years. Institutions have to present a development plan to be later submitted for approval by the Ministry of Education for higher education. The Ministry is responsible for higher education functions as national accrediting diplomas after a favorable opinion is given from the other entities involved. The contract established with institutions and the Ministry has a list of degrees that the educational institution can grant and a commitment by the government to finance the courses for the students enrolled in these programs. The government establishes curricula, program structures, examination regulations, and diplomas. Accreditations are approved after an auditing of the organization of courses and the qualifications and number of faculty members at each school of higher education that has applied for accreditation [38].

### 2.4.9. In India

India's education system originates from the system inherited from Britain in 1947 [39]. India's vast educational system is probably the biggest in the world. The poor quality of higher education has always been a significant problem for all stakeholders in this process. While there is good infrastructure and academic programs, there are problems in other areas and poor performance in research [40]. However, various quality control methods have already been implemented, such as the university affiliation functions or the guidelines of the university grants commission [41]. The National Assessment and Accreditation Council (NAAC) was established in 1994 to ensure and improve quality in higher education and to secure a role at the international level as a quality assurance agency. India intends to continue implementing a learning education program, using best practices, and recognizing institutions with practical methods and features that promote quality in vocational education [42]. This agency has raised awareness and interaction in academia, at various institutions and universities, about course evaluation and accreditation. For 18 years, the NAAC has been well appreciated by academics and institutions for its work as it seeks to evolve its methods and procedures. The accreditation and quality process in

higher education focuses on creating a professional educational system where institutions behave similarly to HR salespeople in matching the national goals of evolution with skills and economic contributions to the competitiveness and development of society. The accreditation and quality process aims to ensure an educational system where excellence in teaching quality is evident, filling gaps and needs in the labor market.

### 2.4.10. In Portugal

The accreditation process is carried out by the Higher Education Evaluation and Accreditation Agency (A3ES) [43]. Higher education courses are evaluated according to the criteria established by the Course Accreditation System (SAC) and, if approved, receive accreditation from A3ES for six years [44]. For the accreditation process of a new course, it is essential to meet all of the criteria established by the SAC and for the institution to promptly provide all required documentation [44]. The accreditation process in Portugal is generally efficient, with most courses receiving accreditation within a reasonable timeframe. However, the authors also note that the process can be more complex and time-consuming for institutions seeking accreditation for the first time or offering new or innovative courses [45].

The National Agency for Quality Assessment and Accreditation (ANECA) carries out the accreditation process in Spain. Higher education courses are evaluated according to the criteria established by the Spanish University Quality Assurance System (SUEQAS) and, if approved, receive accreditation from ANECA for six years. The accreditation process in Spain is generally efficient, with most courses receiving accreditation within a reasonable timeframe. However, the authors also note that the process can be more complex and time-consuming for institutions seeking accreditation for the first time or offering new or innovative courses [46]. The accreditation process of a new course is essential for the course to meet all of the criteria established by SUEQAS and for the institution to provide all required documentation promptly.

The accreditation process for higher education courses can vary significantly between countries, with some countries having more streamlined processes and dedicated accrediting agencies. In contrast, others may have more complex systems with multiple accrediting agencies [47]. In this discussion, we will consider the differences and particularities of the accreditation process in Portugal, Spain, the United States, France, China, Japan, Sweden, the United Kingdom, India, and Germany, focusing on the time and bureaucratic burden of the process and how to speed up the accreditation process of a new course. Table 1 presents relevant results by country.

Regional and national accrediting agencies in the United States carry out the accreditation process. Higher education courses are evaluated according to the criteria established by these agencies and, if approved, receive accreditation for 5–10 years. The accreditation process in the United States is generally efficient, with most courses receiving accreditation within a reasonable timeframe. However, the authors also note that the process can be more complex and time-consuming for institutions seeking accreditation for the first time or offering new or innovative courses [46].

The National Council for Evaluation of Higher Education and Research (CNEAI) carries out the accreditation process in France. Higher education courses are evaluated according to the criteria established by the French Higher Education Quality System (SQAF) and, if approved, receive accreditation from CNEAI for six years. The accreditation process in France is generally efficient, with most courses receiving accreditation within a reasonable timeframe. However, the authors also note that the process can be more complex and time-consuming for institutions seeking accreditation for the first time or offering new or innovative courses [48]. The accreditation process of a new course is important for the course to meet all of the criteria established by the SQAF and for the institution to provide all required documentation promptly.

**Table 1.** Main results by country.

| Country | Results |
| --- | --- |
| China | • The Ministry of Education of the People's Republic of China carries out the accreditation process.<br>• The Chinese University Quality Assurance System (CUQAS) establishes the evaluation criteria for higher education courses.<br>• If a course meets all the criteria and the institution provides all required documentation promptly, it receives accreditation for six years.<br>• The accreditation process is generally efficient but can be complex and time-consuming for institutions seeking accreditation for the first time or offering new or innovative courses. |
| USA | • Accreditation is based on the results achieved in a public process, reflecting whether the higher education institutions meet defined quality objectives.<br>• Innovative methodologies in teaching can lead to better outcomes in course accreditation.<br>• This type of accreditation, unique to education in the United States, compares the performance of educational institutions against metrics defined by the government or accrediting agency.<br>• Certification must be conducted by agencies external to the HEI, and it ensures the institution meets minimum quality requirements.<br>• The accreditation process happens in stages, and the results must be disclosed and published for the institution to receive certification. |
| France | • The government manages the education model in higher education institutions, with some institutions maintaining specific programs.<br>• Higher education institutions (HEIs) have some autonomy in their internal operations, with diverse representation on their boards of directors.<br>• Official accreditation is primarily based on diplomas and degrees, with consideration given to the programs offered by the institution and the resources intended for teaching and training.<br>• The accreditation process is conducted every four years, requiring institutions to present a development plan for approval by the Ministry of Education.<br>• After receiving favorable opinions from various entities, the government accredits diplomas and commits to financing the courses for students enrolled in these programs.<br>• The government also establishes curricula, program structures, examination regulations, and diplomas. Accreditation is approved after auditing the organization of courses and the qualifications of faculty members at each higher education institution. |
| Germany | • Analysis of higher education (HE) assessment exposed a deplorable situation in Germany, leading to decreased public investment in scientific and educational funds.<br>• German students were found to need 3 to 5 additional semesters to obtain their degree compared to other European students.<br>• The Foundation of the Center for the Development of Higher Education was created to promote HE quality at German universities through internal and external evaluations.<br>• There were problems in establishing comparisons between the performances of universities.<br>• The country faced increased HE privatization and legal and economic issues related to new performance-based financing systems.<br>• Reforms were introduced to increase competition and differentiation through deregulation, performance orientation, and performance incentives.<br>• Germany was criticized for not having standards for selecting HE evaluators or evaluating their performance.<br>• There has been a push to widen access to HE by removing the criterion of educational attainment in secondary education as the determining factor for access. |

**Table 1.** *Cont.*

| Country | Results |
|---|---|
| India | • India's education system, the largest in the world, originates from the British system inherited in 1947.<br>• Despite good infrastructure and academic programs, the quality of higher education has been problematic, particularly in research.<br>• Various quality control methods, such as university affiliation functions and guidelines from the University Grants Commission, have been implemented.<br>• The National Assessment and Accreditation Council (NAAC) was established in 1994 to ensure and improve quality in higher education, adopting best practices and recognizing institutions that promote quality in vocational education.<br>• Higher education accreditation and quality process are aimed at creating a professional educational system that aligns with national goals for skills development and economic contributions to societal competitiveness and development.<br>• The accreditation and quality process aims to ensure an educational system where excellence in teaching quality is evident, filling gaps and needs in the labor market. |
| Japan | • Rapid economic development after World War II led to the growth of higher education institutions (HEIs), primarily financed and managed by private companies, which currently run around 80% of HEIs.<br>• Despite considerable variation, the quality of education at these private HEIs is good.<br>• Initially, the Institution Approval System controlled HEIs, but it was set aside in the early 1990s, granting more autonomy to institutions to structure their courses.<br>• New evaluation forms were created, making institutions more directly responsible for quality assurance.<br>• MEXT (Ministry of Education, Culture, Sports Science and Technology) established rigorous parameters and norms to control quality in HEIs in 2001.<br>• Laws were enacted to allow HEIs to self-evaluate and to be evaluated by external companies.<br>• Quality control methods include an approval process where HEIs self-evaluate and control themselves using the CEA process and an evaluation process by the National University Corporation.<br>• Japan's accreditation and evaluation program has become more rigorous to increase accountability and facilitate the implementation of reforms. However, the effectiveness of these measures in making HEIs more competitive and resilient has yet to be established. |
| Portugal | • The Higher Education Evaluation and Accreditation Agency (A3ES) carries out the accreditation process.<br>• Courses are evaluated according to the established criteria of the Course Accreditation System (SAC).<br>• Approved courses receive accreditation from A3ES for six years.<br>• The accreditation process is generally efficient, although it can be more complex and time-consuming for institutions seeking accreditation for the first time or offering new or innovative courses. |
| Spain | • Quality assurance in higher education is a widely discussed topic, with issues such as evaluating student training outcomes, accountability, skills and labor market demands, internationalization and mobility, educational reform, teacher quality, and the impact of economic factors.<br>• The economic recession has restricted some of the monetary resources available to HEIs.<br>• Commitment to establishing and improving the quality of educational systems has been a part of the recent adjustments driven by higher education in Europe.<br>• The European Association for Quality Assurance in Higher Education has published the Standards and Guidelines for Quality Assurance in the European Higher Education Area. |

**Table 1.** *Cont.*

| Country | Results |
|---|---|
| Sweden | • The supervision approach in public administration was implemented in HEIs until the 1980s.<br>• From then on, state control and supervision gradually began to give way to institutional independence and growing self-regulation.<br>• The HEI reform of 1993 decentralized the studies' funds, assignments, and internal assignments.<br>• The National Agency for Higher Education audits and assesses universities based on the national evaluation of subjects and curriculum.<br>• The form of evaluation in enforcement is auditing, which is used to improve and provide ways for the government to audit the quality processes implemented in higher education institutions.<br>• Audits are carried out in three- and four-year cycles, and funding is not linked. |
| United Kingdom | • The Higher Education Funding Council for England (HEFCE) structures the assessment process in England divided into teaching and research assessments.<br>• The results of evaluations must be published for comparison with other HEIs, typically conducted every five to ten years.<br>• Quality is evaluated in terms of return on investment or expenditure associated with accountability.<br>• Research on total quality management (TQM) shows slow adoption in HEIs, but those adopting it show improvements in various aspects.<br>• The business excellence model has demonstrated several advantages in quality management.<br>• Quality assurance in English HE focuses on the attributes of the English system centered on self-regulation and self-governing institutions.<br>• Evaluation of research work is a significant aspect of the academic struggle for quality assurance methods in HEI. |

The Ministry of Education of the People's Republic of China (2003) carries out the accreditation process in China. Higher education courses are evaluated according to the criteria established by the Chinese University Quality Assurance System (CUQAS). If approved, the institution receives accreditation for six years. In the accreditation process of a new course, it is essential to meet all criteria established by CUQAS and for the institution to provide all required documentation promptly. The accreditation process in China is generally efficient, with most courses receiving accreditation within a reasonable timeframe. However, the authors also note that the process can be more complex and time-consuming for institutions seeking accreditation for the first time or offering new or innovative courses [21].

In Japan, the accreditation process is carried out by the Japan Accreditation Board for Engineering Education (JABEE). Higher education courses are evaluated according to the criteria established by the Japanese University Quality Assurance System (JUQAS). If approved, the courses receive accreditation from JABEE for six years [49]. The accreditation process in Japan is generally efficient, with most courses receiving accreditation within a reasonable timeframe. However, the authors also note that the process can be more complex and time-consuming for institutions seeking accreditation for the first time or offering new or innovative courses [49]. To accelerate the accreditation process of a new course, it is important that the course meet all the criteria established by JUQAS and for the institution to provide all the required documentation on time.

The Swedish Higher Education Authority (UKÄ) carries out the accreditation process in Sweden. Higher education courses are evaluated according to the criteria established by the Swedish University Quality Assurance System (SUQAS). If approved, the courses receive accreditation from UKÄ for six years [48]. The accreditation process in Sweden is generally efficient, with most courses receiving accreditation within a reasonable timeframe. However, the authors also note that the process can be more complex and time-consuming for institutions seeking accreditation for the first time or offering new

or innovative courses [48]. To speed up the accreditation process of a new course, it is important for the course to meet all the criteria established by SUQAS and for the institution to provide all required documentation [50] promptly.

The Quality Assurance Agency for Higher Education (QAA) carries out the accreditation process in the United Kingdom. Higher education courses are evaluated according to the criteria established by the UK University Quality Assurance System (UKUQAS). It is important for the course to meet the criteria established by the UKUQAS and for the institution to provide all the required documentation [48]. If approved, it will receive accreditation from QAA for six years. According to one study [51], the accreditation process in the United Kingdom is generally efficient, with most courses receiving accreditation within a reasonable timeframe. However, the authors also note that the process can be more complex and time-consuming for institutions seeking accreditation for the first time or offering new or innovative courses [51].

In India, the accreditation process is carried out by the National Assessment and Accreditation Council (NAAC) [50]. Higher education courses are evaluated according to the criteria established by the Indian University Quality Assurance System (IUQAS) and, if approved, receive accreditation from NAAC for five years. It is important for the course to meet the criteria established by IUQAS and for the institution to provide all required documentation [51]. The accreditation process in India is generally efficient, with most courses receiving accreditation within a reasonable timeframe. The authors also note that the process can be more complex and time-consuming for institutions seeking accreditation for the first time or offering new or innovative courses [52].

In Germany, the accreditation process is carried out by the German Council of Science and Humanities (Wissenschaftsrat) [53]. Higher education courses are evaluated according to the criteria established by the German University Quality Assurance System (GUQAS) and, if approved, receive accreditation from the Wissenschaftsrat for six years [53]. The accreditation process in Germany is generally efficient, with most courses receiving accreditation within a reasonable timeframe. However, the authors also note that the process can be more complex and time-consuming for institutions seeking accreditation for the first time or offering new or innovative courses [54]. It is important for the course to meet all of the criteria established by GUQAS and for the institution to provide all required documentation [53].

It is worth noting that the time and bureaucratic burden of the accreditation process may also depend on the specific accrediting agency and the course offered. For example, some accrediting agencies may have more stringent criteria or require more documentation than others, which could result in a longer or more complex accreditation process [55]. Similarly, some courses, particularly those that are more specialized or innovative, may require more extensive review and evaluation by the accrediting agency [56]. Institutions seeking to speed up the accreditation process of a new course may want to consider these factors and work closely with the relevant accrediting agency to ensure all necessary criteria are met and documentation is provided promptly.

The analysis of the accreditation processes in various countries highlights common trends in pursuing quality assurance in higher education. While there are differences in the criteria, evaluation methods, and timeframes for accreditation, many countries have adopted a systematic approach to quality assurance to promote transparency, competition, and customer satisfaction.

## 3. Results and Discussion

Comparing different countries' accreditation processes reveals a general trend toward efficiency, with most institutions receiving accreditation within a reasonable timeframe. However, institutions seeking accreditation for the first time or offering new or innovative courses face greater challenges, suggesting that accrediting agencies may need to reconsider their criteria and processes for such cases.

Additionally, the study identifies a shift in power within higher education, with institutional levels benefiting from quality assurance practices and extrinsic values being reinforced at the expense of intrinsic values. Managerial concerns such as those of the market have become more important than have academic disciplinary concerns.

### 3.1. Implications for Theory

The comprehensive analysis of accreditation processes worldwide enhances our understanding of quality assurance in higher education. It emphasizes the need for theoretical frameworks accommodating diverse practices across countries and contexts. The observed shift of power within higher education and the increasing importance of market concerns contribute to the growing body of theory on the marketization of education and its implications for academic integrity and institutional autonomy.

### 3.2. Implications for Practice

For practitioners, this research offers valuable insights into the accreditation practices in different countries, allowing institutions to benchmark their processes and understand the criteria they need to meet. The study highlights the importance of transparency in quality assurance, informing policy-making in higher education and encouraging governments to implement regulations promoting transparency and competition. The observation that accreditation can be more challenging for institutions offering new or innovative courses calls for accrediting agencies to adapt their criteria and processes to accommodate such courses and avoid stifling innovation in higher education.

### 3.3. Implications for Future Research

This research opens several avenues for future exploration, such as investigating the specific challenges faced by institutions offering new or innovative courses in the accreditation process and identifying ways to overcome these challenges. Future studies could also examine the implications of the observed shift in power within higher education for the quality of education and student outcomes. Moreover, the growing importance of market concerns in higher education prompts questions about their impact on educational quality, accessibility, and equity. Future research could explore these issues in depth, informing both theory and practice in higher education.

Lastly, a longitudinal study tracking the changes in accreditation processes and criteria over time could provide insights into the evolution of quality assurance in higher education in response to changing societal and market demands.

Quality in higher education is complex, as it can mean different things to different people. Some authors argue that there are five primary ways that "quality" is understood: as exceptional (high standards), as perfection (consistency), as fitness for purpose (meeting a specified requirement), as value for money (cost-effectiveness), and as transformative (changing for the better) [57].

Quality assurance in higher education involves measures to ensure that standards are being met and that the quality of learning is being continually improved. There is considerable variation globally in how quality assurance is understood and implemented. Vlăsceanu, Grünberg, and Pârlea [58] detail the differing approaches across various countries, noting that these can be broadly grouped into external quality assurance (external audits or accreditation) and internal quality assurance (self-evaluation or internal review processes).

A key discussion point in the literature revolves around whether quality assurance should be viewed as a process of control or improvement. Some authors suggest that quality assurance has historically focused on accountability and control, emphasizing external reviews and meeting predetermined standards [59]. On the other hand, others like Dill [60] have called for a shift toward improvement-oriented quality assurance, which focuses on enhancing learning and teaching practices.

Another significant theme in the literature concerns the tensions between standardization and diversity. Quality assurance processes focusing on standardization can undermine the diversity and creativity inherent in higher education. However, authors like Van Damme note that standardization is necessary to ensure comparability and consistency [61].

Stensaker and Harvey have explored the impacts of quality assurance on universities, highlighting both intended and unintended consequences. While quality assurance can lead to improved practices and increased accountability, it can also result in compliance culture and "gaming" behaviors where universities seek to manipulate or manage performance indicators rather than focus on genuine improvement [62].

Looking at the future of quality assurance, Ewell argues for a shift toward a learner-centered approach to quality, focusing on learning outcomes and the student experience [63]. Meanwhile, Hazelkorn discusses the need for quality assurance to adapt to the changing landscape of higher education, including the rise of online learning, transnational education, and the increasing diversity of learners [64].

Overall, the literature highlights the multifaceted nature of quality assurance in higher education and points to ongoing debates about its purpose, focus, and impacts. The complexities of defining and assuring quality in the diverse and rapidly evolving context of higher education continue to present significant challenges and opportunities.

## 4. Conclusions

This study delved into the intricate accreditation and quality assurance process in higher education across multiple countries, offering a comprehensive overview of the practices worldwide. Common trends were observed, such as a systematic approach to quality assurance, the pursuit of transparency, and a drive for competitiveness. Despite the differences in criteria, evaluation methods, and timeframes, these shared characteristics underscore a universal endeavor to enhance customer satisfaction in the educational landscape. These research findings also signal a notable shift in power dynamics within higher education, with increased leverage at the institutional level and a noticeable emphasis on extrinsic values over intrinsic ones. This power realignment and the prioritization of market factors have far-reaching implications for the theoretical understanding of the marketization of education and the practical issues concerning academic integrity and institutional autonomy. Intriguingly, market-related concerns appeared to overshadow academic disciplinary concerns in this scenario. While accreditation processes typically display efficiency, it was identified that first-time institutions or those offering innovative courses must often encounter more complexity and time consumption. This insight encourages accrediting bodies to revisit and adjust their criteria and processes, ensuring they foster innovation rather than impede it. The study has significantly enriched the understanding of higher education quality assurance and revealed new areas for future research. Future explorations could delve into challenges faced by innovative institutions in the accreditation process, the implications of the power shift within higher education, and the impact of market concerns on educational quality, accessibility, and equity. Further, a longitudinal study could offer valuable insights into how quality assurance evolves in response to societal and market demands.

In conclusion, this study underscores the pivotal role of accreditation in maintaining and enhancing the quality of higher education. As the education sector evolves and adapts to societal changes so must the processes guaranteeing its quality and relevance. Understanding the mechanisms of accreditation and their impact on educational institutions is thus crucial for all stakeholders, from policymakers to educators, students, and their families.

**Author Contributions:** Conceptualization, N.D. and R.V.; methodology, N.D. and R.V.; investigation, N.D. and R.V.; writing—original draft preparation, N.D.; writing—review and editing, R.V. All authors have read and agreed to the published version of the manuscript.

**Funding:** The research was partially funded by ISLA Santarém, who supported the APC.

**Conflicts of Interest:** The authors declare no conflict of interest.

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
