# Peer review of "Literature Review of Accreditation Systems in Higher Education"

_education, doi:10.3390/educsci13060582_

Round 1

Reviewer 1 Report

It was a pleasure to review this manuscript on accreditation systems in Higher Education. Many areas of strength stand out in this paper. However, I would like to focus on areas of growth as below:

The title of the manuscript needs to be revised because the text discusses the accreditation systems in higher education, Not Course Accreditation.

The abstract section needs to include a general statement, aim/purpose, methods, key findings, and implications. At this point, the manuscript lacks the final two elements.

In the methods section, readers would be curious to know about the selection criteria for articles since this is a literature review. Perhaps you can present a conceptual framework that guides the selection of articles, and regions.

You have presented Harvey and Green’s framework defining quality in higher education, but there is no link to the text explaining how your analysis is linked to the framework.

The literature you present focuses on both institutional and program accreditation. In the introduction section, you better explain the scope of accreditation and the rationale for why your selection criteria include both institutional and programmatic accreditation. Given that you never talk about courses throughout the paper, you may need to think about how to revise the title to avoid confusion.

I like the way you present each country’s cases; however, your analysis should synthesize common trends between various countries by comparing and contrasting them. More specifically, your discussion section should include your analysis of the literature with the conceptual framework, and discuss key implications for theory, practice, and future research. In other words, your manuscript needs to go beyond an annotated bibliography and weave the pieces together.

Overall, the quality of English language is great. However, there are some word choices that need to be fixed in order to publish in a journal article.

Author Response

The authors would like to thank the reviewer interest and time in review their manuscript suggesting important suggestions and raising comments to improve its quality.

Reviewer #1:

It was a pleasure to review this manuscript on accreditation systems in Higher Education. Many areas of strength stand out in this paper. However, I would like to focus on areas of growth as below:

The title of the manuscript needs to be revised because the text discusses the accreditation systems in higher education, Not Course Accreditation.

R: The authors would like to thank the reviewer kind words towards this manuscript, it is a huge incentive for researching on this subject. The manuscript title was adjusted as suggested.

The abstract section needs to include a general statement, aim/purpose, methods, key findings, and implications. At this point, the manuscript lacks the final two elements.

R: The abstract was rewritten and made more complete. It accommodates a general statement, aim, methods, key findings and implication.

In the methods section, readers would be curious to know about the selection criteria for articles since this is a literature review. Perhaps you can present a conceptual framework that guides the selection of articles, and regions.

R: We thank the reviewer for this comment, the bibliography selection criteria was added to the new materials and methods section.

You have presented Harvey and Green’s framework defining quality in higher education, but there is no link to the text explaining how your analysis is linked to the framework.

R: Thanks again for spotting this, a whole paragraph on how Harvey and Green’s framework defining quality in higher education is linked to our research was added to the new materials and methods section.

The literature you present focuses on both institutional and program accreditation. In the introduction section, you better explain the scope of accreditation and the rationale for why your selection criteria include both institutional and programmatic accreditation. Given that you never talk about courses throughout the paper, you may need to think about how to revise the title to avoid confusion.

R: we are very grateful for this suggestion to improve our paper, the introduction was completely rewritten for better explaining the scope of accreditation and the literature selection criteria rationale, and as mentioned before the title was changed upon on your suggestion to avoid any confusion.

I like the way you present each country’s cases; however, your analysis should synthesize common trends between various countries by comparing and contrasting them. More specifically, your discussion section should include your analysis of the literature with the conceptual framework, and discuss key implications for theory, practice, and future research. In other words, your manuscript needs to go beyond an annotated bibliography and weave the pieces together.

R: Thanks, one more time for raising this, a results and discussion section was added comparing the literature findings and the table 1 was also improved. This new section helped to linking all aspects of the manuscript together making it more robust and improving its quality.

We are very grateful for all the suggestions and comments, which we thing allowed to improve greatly our research paper.

Reviewer 2 Report

The paper titled "Literature Review of Course Accreditation Systems in Higher Education" is reviewed. A number of related recommendations are presented as follows:

·       The current abstract is too short.

·        The authors should first define abbreviations before using them.

·       In addition, The authors might use an abbreviation table at the beginning or end of the text since lots of abbreviations are used in the text.

·       The information in subheading 2.2 should be reviewed, it is not clear enough.

·       It should be added that the countries determined for the method of the research were determined according to which criteria.

·       What does the sentence "Error reference source not found" mean?

·       On lines 521 and 522 it was written that Future 521 work will focus on a paper in the following order in the accreditation of a course. However, no ranking was mentioned.

·       The authors did not mention the suggestion part in their studies.

Author Response

The authors would like to thank the reviewer interest and time in review their manuscript suggesting important suggestions and raising comments to improve its quality.

A number of related recommendations are presented as follows:

  • The current abstract is too short.

R: the abstract was rewritten and made more complete.

  • The authors should first define abbreviations before using them.

R: All abbreviations were evaluated and at their first occurrence were defined.

  • In addition, The authors might use an abbreviation table at the beginning or end of the text since lots of abbreviations are used in the text.

R: The authors decided to define the abbreviation at its first occurrence in the text, although we are thankful to the reviewer to suggest a table at the beginning or end of the manuscript.

  • The information in subheading 2.2 should be reviewed, it is not clear enough.

R: The whole text of section 2.2 was rewritten and made clearer and objective as suggested by reviewers.

  • It should be added that the countries determined for the method of the research were determined according to which criteria.

R: We are thankful for this suggestion, which was addressed with the addition of the text: “The techniques employed across Europe, the United States, and Asia were researched based on their diverse approaches to higher education quality assurance, their significant influence on global education policies, and their varied cultural and administrative contexts.”

  • What does the sentence "Error reference source not found" mean?

R: Thank you for pointing this out, it was a bibliographic reference missing and it was corrected.

  • On lines 521 and 522 it was written that Future 521 work will focus on a paper in the following order in the accreditation of a course. However, no ranking was mentioned.

R: The authors thank the reviewer for pointing this and decided to rewrite the further research direction differently.

  • The authors did not mention the suggestion part in their studies.

R: Suggestions were added in the two last paragraphs of the revised version of the manuscript.

We are very grateful for all the suggestions and comments.

Reviewer 3 Report

1. Most of the references are out dated

2. Inconsistent in citation style

3. Remove irrelevant information  

4. Some information in Table 1 did not match with the objectives of the paper  

Author Response

The authors would like to thank the reviewer interest and time in review their manuscript suggesting important suggestions and raising comments to improve its quality.

  1. Most of the references are out dated

R: We thank this comment but given the stated defined eligibility criteria we were unable to find any recent literature references fallen on it, but if the reviewer has found any, we would be thankful if he can provide us the details and we will consider including them after careful consideration.

  1. Inconsistent in citation style

R: The citation style was reviewed and adequately addressed.

  1. Remove irrelevant information  

R: All the reviewers pointed some irrelevant and repeated information in the original manuscript, authors are thankful for that and  addressed their concerns, in all sections.

  1. Some information in Table 1 did not match with the objectives of the paper  

R: We are very thankful for this comment, the table 1 was evaluated and corrected accordingly, matching the objectives of the manuscript.

We are very grateful for all the suggestions and comments.

Round 2

Reviewer 1 Report

Thank you for letting me review your paper for the second time. I see significant improvements in your paper compared to the first version. My main suggestions are related to English language use. For example, words such as guarantee, critical have strong meanings that may not be the good word choice when introducing accreditation. 

Accreditation is an important process that assesses a program or institution against benchmarks that inform quality. However, one cannot claim that accreditation guarantees quality. 

Overall, the manuscript is in good standing. I would suggest expanding the discussion section by providing more evidence from the literature as opposed to generic comments.

The manuscripts needs minor editing, particularly, to avoid the use of judgmental words. 

Author Response

The authors would like to thank the reviewer interest and time in review their manuscript suggesting important suggestions and raising comments to improve its quality.

Thank you for letting me review your paper for the second time. I see significant improvements in your paper compared to the first version. My main suggestions are related to English language use. For example, words such as guarantee, critical have strong meanings that may not be the good word choice when introducing accreditation.

R: We would like to thank the reviewer to point this out and the English Language use was reviewed appropriately.

Accreditation is an important process that assesses a program or institution against benchmarks that inform quality. However, one cannot claim that accreditation guarantees quality.

R: The claim of accreditation guarantee quality was replaced by accreditation supports quality.

Overall, the manuscript is in good standing. I would suggest expanding the discussion section by providing more evidence from the literature as opposed to generic comments.

R: The discussion section was improved as recommended.